# Dependence Models of Borehole Expansion on Explosive Charge in Spherical Cavity Blasting

**Denis Težak** [1], **Siniša Stanković** [2,*] and **Ivan Kovač** [1]

1   Faculty of Geotechnical Engineering, University of Zagreb, Hallerova aleja 7, Varaždin 42000, Croatia
2   Faculty of Mining, Geology and Petroleum Engineering, University of Zagreb, Pierottijeva 6,
    Zagreb 10000, Croatia
*   Correspondence: sinisa.stankovic@rgn.hr

**Abstract:** In geotechnical practice, it is often necessary to improve the properties of soil and rock in which different structures are built. For this purpose, spherical cavity blasting can be applied to expand the borehole. Such expansion may incorporate various constructive elements such as anchors and thus stabilize the slope. The paper presents the method for determining the increased volume, expansion, and deepening of the borehole as a result of spherical cavity blasting. In addition, mathematical models describing the dependency of the borehole expansion on the amount of explosive charge are presented. The models are mutually compared with the Akaike information criterion.

**Keywords:** spherical cavity blasting; borehole; explosive loading; expansion dependency model

## 1. Introduction

The spherical cavity blasting method in clay soils gives many positive effects in terms of improving the properties of clay in a wider area of explosive activity and creating a local spherical cavity at the bottom of the borehole. The possibilities of the application and the effects of this kind of blasting have not been sufficiently investigated, especially in relation to the simplicity of the method and the level of improvement of the soil properties and the increase in the volume of the borehole.

The term "spherical cavity blasting" basically explains the borehole enlargement which is achieved by a one-time or successive repetition blasting of a small quantity of explosive located at the bottom of the borehole [1]. The main explosive charge is in the borehole expansion area, and the secondary explosive charge is in the borehole right above the main charge. After the initial blasting, detonation gas pressure interacts with the soil/material on the bottom of the borehole [2]. As a result, the ground in the zone around explosion begins to move and the distant layers are compressed [3], creating spherical cavity. The formation of spherical cavity expansion by successive repetition blasting in phases is shown in Figure 1.

The main effects of blasting in soils, particularly clay, are shown in the installation of the structural elements for anchoring of foundations and retaining walls, permanent stabilization of clay slope, and stabilization of various commercial structures such as overhead transmission towers, tunnels, etc.

The example of the application of the method is compacting coherent clay soil by blasting for slope stabilization by anchoring for different surface and underground structures. By the detonation of the generated shock wave and the detonation gas pressure in the soil, a pore overpressure and intense oscillations are created in the vicinity of the explosive charge. During this process, the natural structure of the clay is destroyed, the free volume (cavity) is created near the explosive charge, and in the zone along the borehole wall, i.e., the enlargement, leads to intense compression of the clay with increasing density (Figure 2).

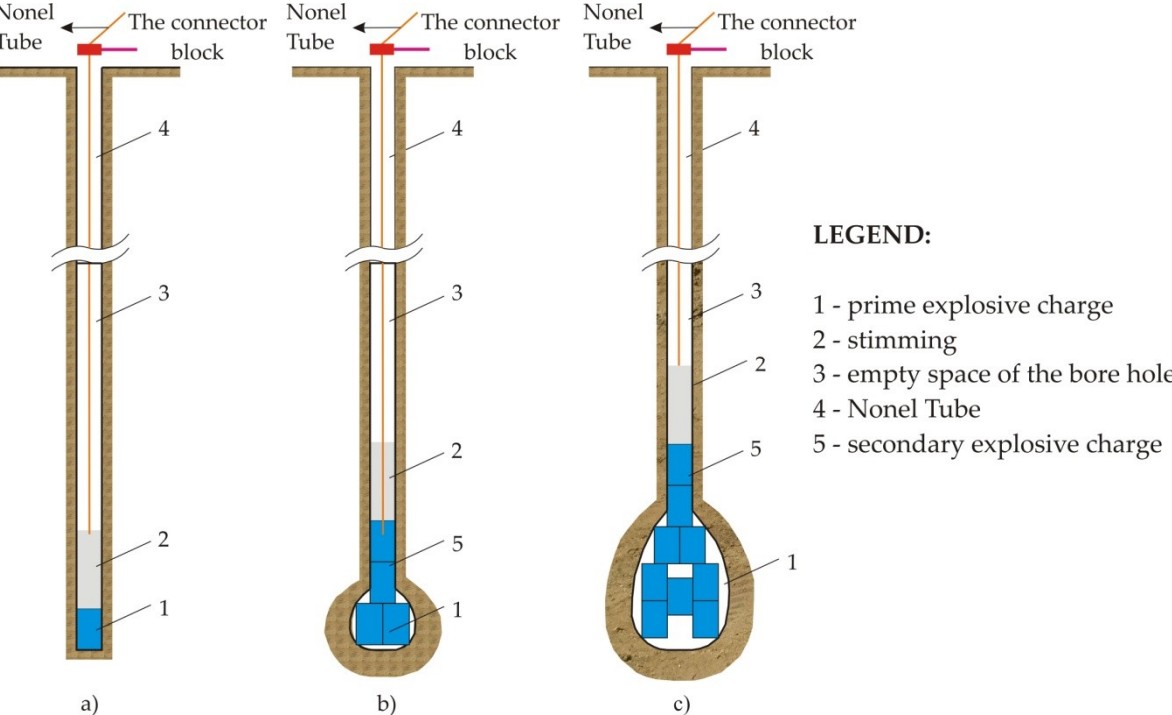

**Figure 1.** Spherical cavity blasting phases (**a**) Before the detonation of the explosive charge; (**b**) After detonation of the explosive charge; (**c**) Charging the spherical cavity of the borehole [1].

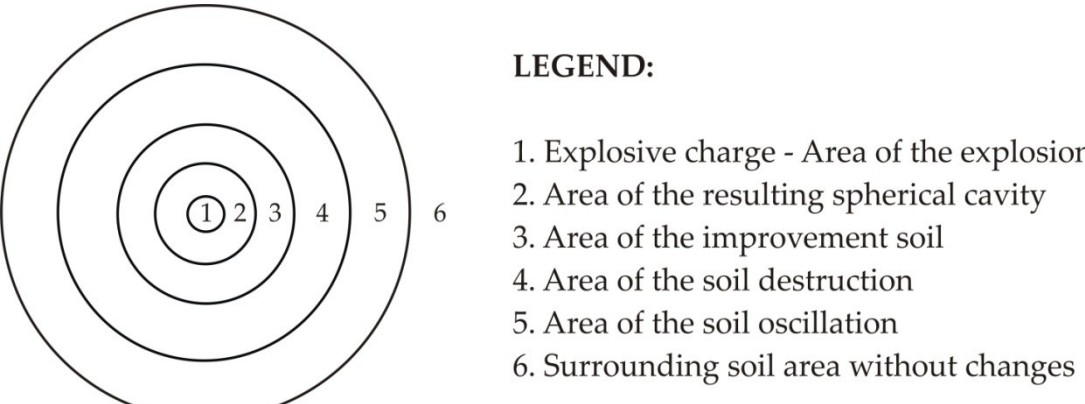

**Figure 2.** Performance of explosion in soft rock.

A measurement of success of blasting is spherical expansion and the volume of the generated expansion (Figure 3). The extent of expansion and properties of the compressed clay can be determined by field and laboratory methods. Optimization of spherical cavity blasting in clay can be achieved from the aspect of explosive type and mass impact, blasting parameters and reduction of potentially harmful environmental effects. Determination of the influence and effects of certain explosive charges can be carried out by the original method of recording and measuring the resulting spherical cavity expansion, developed for the purposes of the above-mentioned research.

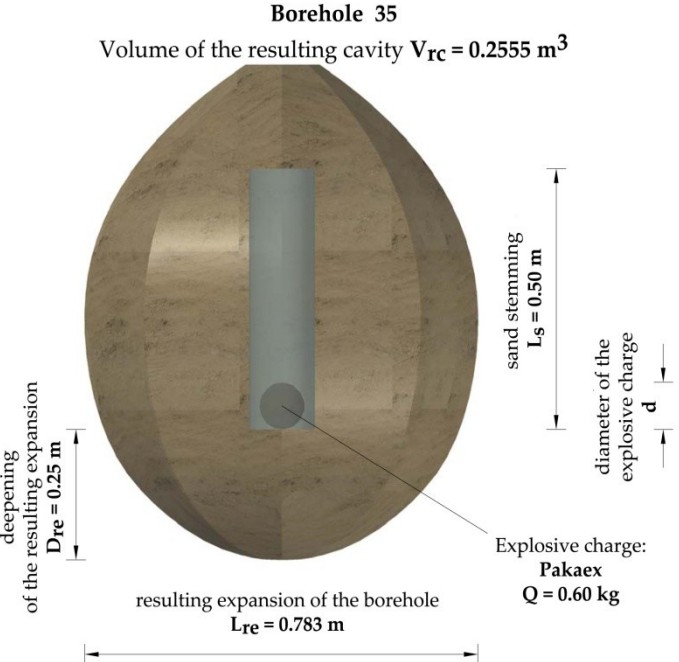

**Figure 3.** Formed spherical cavity expansion in 3D.

For a complete analysis of the effects of spherical cavity blasting, it is necessary to create mathematical models that describe the dependence of increased volume, expansion, and deepening of the borehole on the amount of explosive charge. The results obtained by this research are graphically presented in the scatter plots. The results are shown as regression curves of different models. The reliability of the model and the success of their integration into experimental data has been established.

The paper presents the results of spherical cavity blasting for three blasting performance indicators: increase in volume, expansion, and deepening of the blasthole. Two types of explosives were used during the research: Pakaex and Permonex, with following properties, Pakaex: density 0.87 g/cm$^3$, VOD 2950 m/s, gas volume 984 l/kg, and Permonex: density 0.95 g/cm$^3$, VOD 4500 m/s, gas volume 900 l/kg. Therefore, six cases "indicator-explosive" are defined in this way.

This study has two goals:

- Presenting a new method for determining the value of spherical cavity blasting performance indicators;
- Determining the mathematical model that best describes the dependence of blasting results on the amount of explosive charge.

## 2. Previous Research

In the late eighties, some research was carried out to determine the possibilities of extending anchors and braces to structures constructed in soft soil [4]. Excavation through earthwork is one of the most complex tasks in the construction of underground structures and tunnels. Therefore, attempts have been made to develop procedures that can facilitate the creation of underground structures in rock that are, by their geotechnical characteristics, considered soils. Almost all well-known anchoring procedures by means of reinforcement bars, pipes, or prestressing cables, using cement or plastic injection mixes have not been sufficiently successful in anchoring made in clay, loam, or similar soft or earthen materials. The anchors are performed in the form of rigid profiles (reinforced iron bars or pipes), or in the form of steel cables [5]. This patent-protected process is based on the fact that the detonation of a given mass of explosives, located in the coherent ground borehole, results in a limited expansion most commonly of a spherical shape. The volume of the sphere depends on the mass and type of the explosive used and of course on the geotechnical characteristics of the soil [6–8].

In addition to the usual construction methods that effectively improve the geotechnical properties of soft soil in the wider area, in some cases it is possible to use the energy released by the detonation of the explosive charge below the soil surface, called explosive compaction. The explosive compaction (EC) is basically a soil modification technique by which detonation-free energy of explosives in the underground is used to compact the soft soil layers. The effectiveness of the EC depends mostly on the soil profile, soil particle size distribution, and type of explosive. By analyzing the results of field trials carried out on 13 sites worldwide, where EC has been successfully applied to soil modification, it has been established that the EC can be an effective method for improving the density, stability, and strength of clay. The results of the application showed a suitable efficiency for unstable and liquefiable deposits having particle sizes ranging from gravel to dry sand with less than 10% clay content. [9].

In addition to the above, according to available literature sources, it can be concluded that improvement of the geotechnical properties of soft soil by the explosive compaction method (EC) is utilized to a lesser extent than using conventional construction methods, such as: "vibratory methods, deep soil mixing (DSM), deep dynamic compaction (DDC), or jet grouting".

However, the literature shows developed method and 3D visualization for displaying the natural discontinuity in the rock within drilled boreholes. The application of this method is 3D visualizations of fracture distribution in volumes close to the boreholes for well planning, reservoir-scale fracture model building, reservoir flow simulation, and hydraulic fracture control [10].

The classification and characterization of rock structures have been defined by a combined interpretation of core images and acoustic images of borehole wall [11].

The BoreIS (software) was developed as an extension to the ESRI Arcscenes three-dimensional (3D) GIS environments. BoreIS' interactive manipulation of terms in complex queries, simple addition of contour surfaces, and masking by lithology or formation helps geologists find spatial patterns in their data, beyond the limits of data tables and flat maps [12].

Following the above, there was a need to develop an application and a unique method for calculating the volume and 3D display of the resulting spherical expansion. The described method shows the integration of GNSS data measurements and data obtained by measuring with borehole camera and laser.

## 3. Recent Research

The most recent field researches were carried out during 2014, 2015, and 2016 at clay pit Cukavec II, near Varaždin, [13] (Figure 4). The observing instruments for reading field data and program support that analyzed and interpreted the results of the research were used.

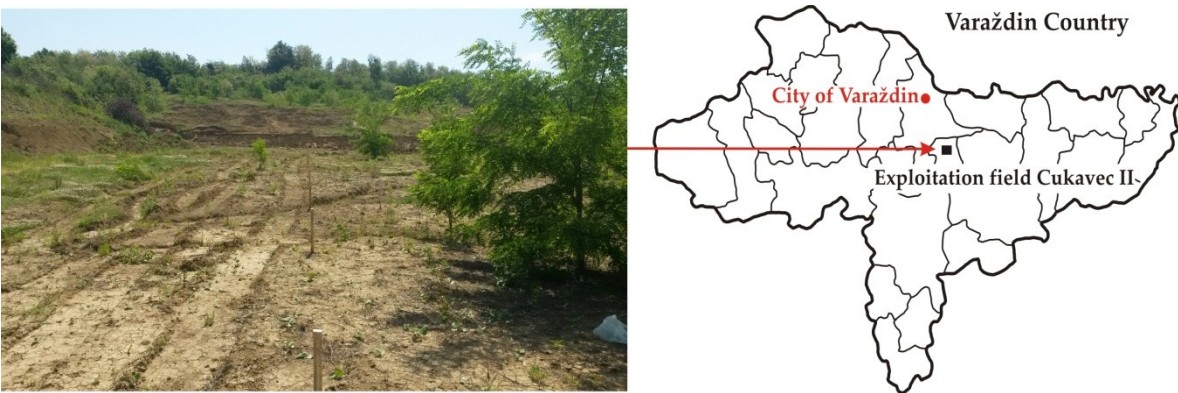

**Figure 4.** Location of the exploitation field Cukavec II [14].

Expansions resulting from the spherical cavity blasting in Cukavec II clay are different in form and size. In homogeneous clay soils it is formed in the spherical shape [15]. The spherical expansions of boreholes were created by the detonation of the Ammonia Nitrogen Powder Explosive Permonex

V19 and ANFO explosives commercially known as Pakaex. The velocity of detonation is a detonation parameter that can be used to determine the performance of used explosive in different rock types also in a coherent clay.

During the research, of particular importance was the determination of the shape and volume of the expansions formed at a certain depth of clay soil after the blasting of explosive in a cylindrical borehole with a diameter of at least 131 mm. To this end, the system for observation, measurement, and calculation of the volume of the resulting enlargement was developed at the Faculty of Geotehnical Engineering. This system represents the integration of the RTK GNSS method, using online transformation parameters via CROPOS [16,17], depth cameras; Heavy Duty Geo Vision Borehole Camera [18] and laser, EDS-C [19] (Figure 5).

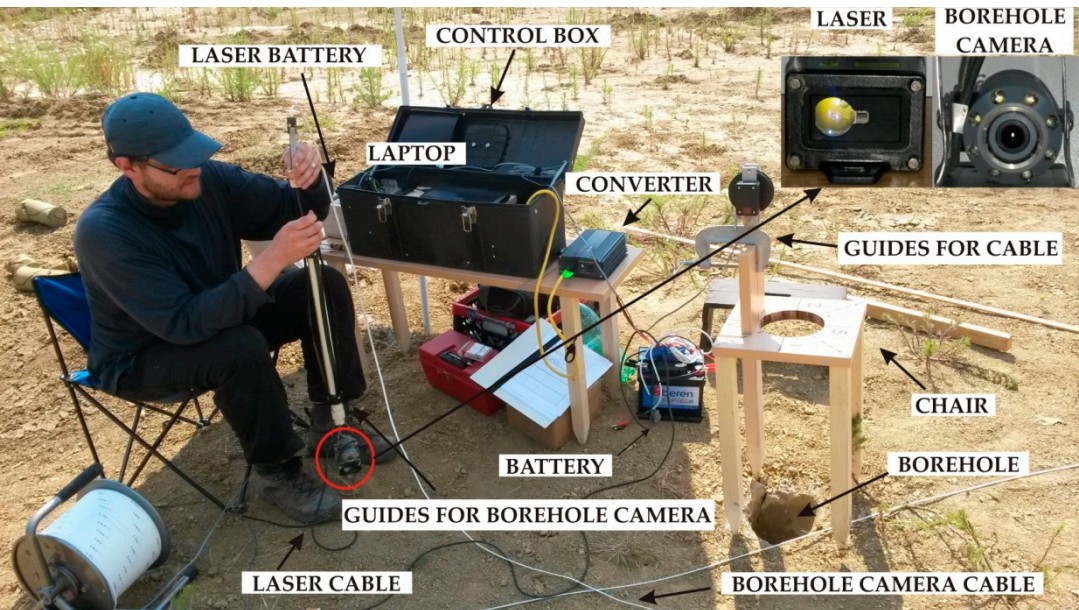

**Figure 5.** Spherical expansion measurement equipment [14].

Also, the software "Bušotine" (Boreholes) was used for calculation of borehole coordinates based on the known $E$ and $N$ coordinates (obtained by the GPS device) and the height $H$ (in this case of the borehole depth) obtained using the depth camera and the laser. For the calculation of the resulting expansion and its volume at a certain depth, an application has been used which, besides coordinates calculation, gives a detailed graphical interpretation in the 2D and 3D view [14]. Scrutinizing available literature, no researches were found in which a method for determining shape and volume of spherical expansion was defined.

A conceptual plan and methodology of field research has been developed. The field research plan was divided into:

- **Locating blasting areas and positioning of the future boreholes**

  - In order to determine the borehole position coordinates and height, is the determination of the geodetic profile (Figure 6), was performed with the RTK GNSS method, using online transformation parameters via CROPOS (CROatianPOsitioning System).

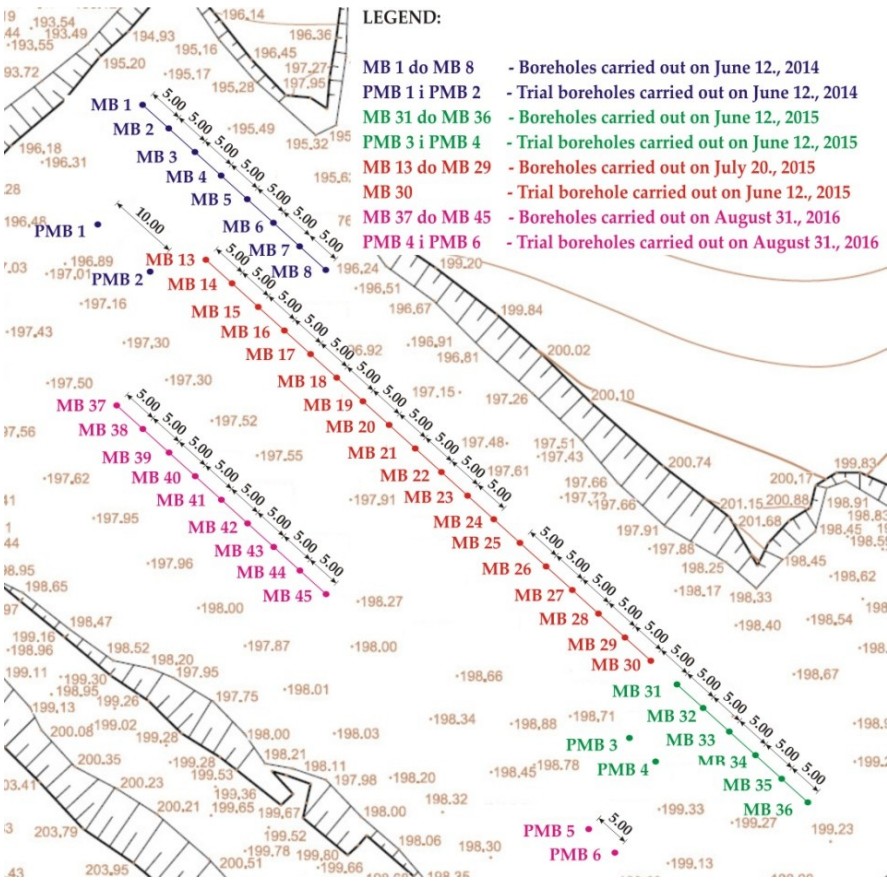

**Figure 6.** Exploitation field Cukavec II and positioned boreholes.

- **Geotechnical field investigations**

  - For the assessment of the dynamic properties of the soil at the profile depth before and after the blasting, the MASW (multi-channel analysis of surface waves) method for multichannel analysis of surface waves, have been used for measurement of velocity of the shear waves $V_s$ [20,21]. The aim of the geophysical research was to determine the data on the general and mechanical properties of the soil natural layers by depth, and to determine the changes of the soil dynamic properties caused by the activation of explosive charges of different type and masses;

  - Exploratory drilling of boreholes was carried out with the aim of obtaining disturbed and undisturbed samples of clay, which were taken from characteristic boreholes at certain depth, and sent to geotechnical laboratory for further testing;

  - Geomechanical laboratory tests of disturbed and undisturbed samples of the subject clay have been performed in an accredited laboratory of the Faculty of Geotehnical Engineering according to the international standard HRN EN ISO / IEC 17025: 2007. For the purposes of research, the clay moisture is determined, with its undrained shear strength before and after blasting.

- **Test blasting and determination of the effective range of masses of two different types of explosives**

  - It was found that the spherical expansion for the predetermined borehole diameter of 131 mm and the depth from 2.00 to 3.00 m, is possible with the explosive type Permonex V19 and Pakaex, ranging from 0.2 to 1.6 kg. Declared velocity of detonation for Permonex

V19 explosive is 4500 m/s, while for the Pakaex explosive is 2950 m/s. The explosive charge activation was performed by the NONEL system started with the instantaneous electric detonators (IED) (Figure 7). During the test blasts, the stem was 0.3 m and 0.5 m, depending on the quantity of explosive in the borehole, not to allow its ejection from it, which would result in partial detonation energy loss. The stone material gradation 0/4 mm was used for stemming. Figure 8a shows the borehole construction. The moment of explosive charge activation of the Permonex V19 is shown in Figure 8b, and the schematic representation of the resulting spherical expansion is shown in Figure 8c [14].

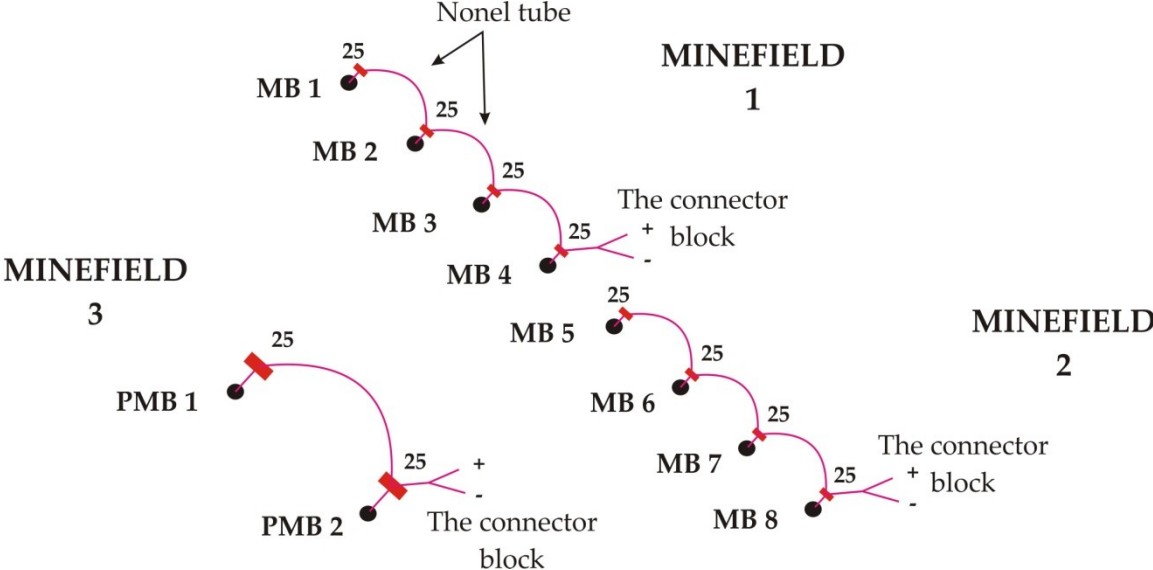

**Figure 7.** Schematics of blasting area with position of boreholes.

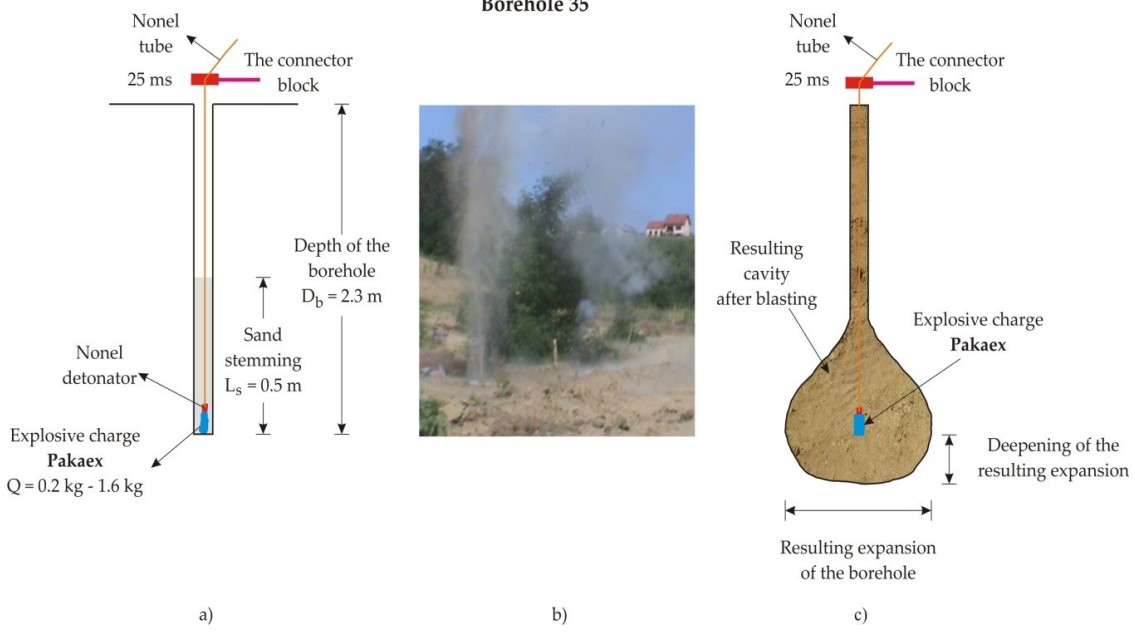

**Figure 8.** (**a**) Borehole construction;(**b**) Moment of explosive charge activation; (**c**) Formed spherical expansion.

- **Determination of spherical expansion after explosive charge activation**

-   The most important was the determination of the shape and volume of the spherical expansions formed after the explosive charge detonation in a 131 mm in diameter cylindrical borehole.

## 4. Spherical Expansion Volume Measuring Method

The system consists of a depth camera, a laser, a control box, a laptop, a battery, an inverter, two 12 V portable batteries for laser, and cables (Figure 5).

Depth camera, connected with cable, is attached to a camera bracket connected to the control box. The laser is secured to the bottom of the depth camera with the help of a bracket and connected with the cable to the control box. Another cable connects the laser to two 12V batteries that enable functioning of the laser, and to a laptop with a special converter. The control box is connected to a 55Ah battery using the cable and inverter. A laptop using software for the recording and data storage is connected to the control box and allows the start of the measurement. Depth camera and the laser are attached along with bracket to the cable guide, through the circular opening on the chair and lowered into the borehole. Depth camera rotates 360° clockwise at the typical recording depths, with 45° intervals. This way the closed surface is obtained. Since the laser is attached to the depth camera, the distance of the depth camera from the borehole walls is recorded in each second [14].

Using a laptop, a laser and a camera, and laser software, a video file of the entire borehole is recorded along with the distances of the camera from borehole wall (Figure 9) [14].

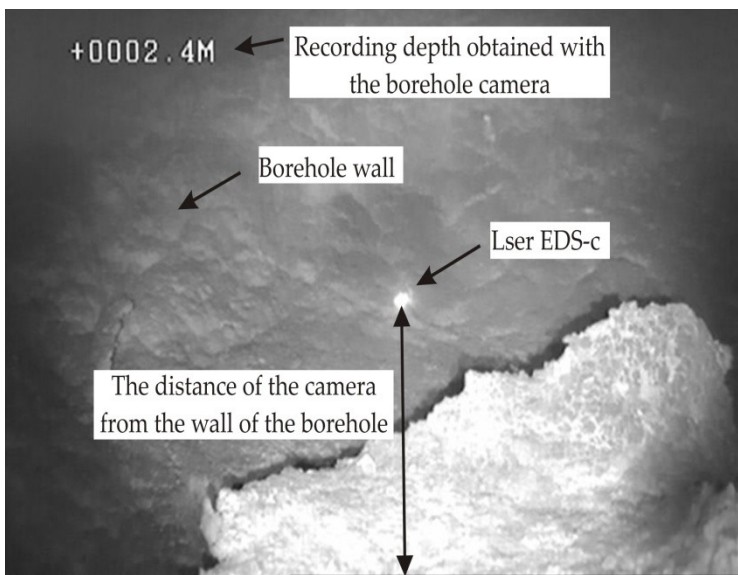

**Figure 9.** Depth camera snapshot of the spherical expansion.

## 5. Models

Based on the data displayed in Table 1, following scatter plots were formed: The quantity of explosives (Q)–volume of expansion ($V_{rc}$); the quantity of explosives–diameter of the expansion ($L_{re}$) and the quantity of explosives–depth of expansion ($D_{re}$) for both types of explosive. In the scatter plots, regression curves (exponential, logarithmic and power) were implemented, which represent dependence models of increased volume, expansion, and deepening of the borehole on the amount of explosive charge, Table 2.

**Table 1.** The results of spherical cavity blasting for explosives Pakaex and Permonex V19.

| | Pakaex | | | | | Permonex V19 | | | |
|---|---|---|---|---|---|---|---|---|---|
| Borehole | Explosive Charge Mass | Volume of the Resulting Cavity | Resulting Expansion of the Borehole | Deepening of the Resulting Expansion | Borehole | Explosive Charge Mass | Volume of the Resulting Cavity | Resulting Expansion of the Borehole | Deepening of the Resulting Expansion |
| | $Q$ (kg) | $V_{rc}$ (m³) | $L_{re}$ (m) | $D_{re}$ (m) | | $Q$ (kg) | $V_{rc}$ (m³) | $L_{re}$ (m) | $D_{re}$ (m) |
| MB20 | 1.00 | 0.7100 | 1.1570 | 0.5200 | MB24 | 0.80 | 0.6184 | 1.1900 | 0.3100 |
| MB41 | 1.00 | 0.8095 | 1.1110 | 0.6000 | MB26 | 0.80 | 0.5690 | 1.1310 | 0.3600 |
| MB34 | 0.80 | 0.3935 | 0.9530 | 0.3300 | MB45 | 0.80 | 0.7405 | 1.0700 | 0.4000 |
| MB18 | 0.80 | 0.3440 | 0.8770 | 0.4600 | PMB5 | 0.80 | 0.7227 | 1.0710 | 0.4200 |
| MB19 | 0.80 | 0.3626 | 0.8750 | 0.4800 | MB23 | 0.60 | 0.5276 | 1.1040 | 0.3500 |
| MB40 | 0.80 | 0.5190 | 1.0600 | 0.4000 | MB25 | 0.60 | 0.6330 | 1.0850 | 0.2900 |
| MB35 | 0.60 | 0.2555 | 0.7830 | 0.2500 | PMB6 | 0.60 | 0.6151 | 1.1520 | 0.3500 |
| MB17 | 0.60 | 0.6160 | 1.0430 | 0.3400 | MB36 | 0.40 | 0.1135 | 0.6930 | 0.2300 |
| MB39 | 0.60 | 0.3785 | 1.0880 | 0.4000 | MB21 | 0.40 | 0.2925 | 0.9360 | 0.2600 |
| MB15 | 0.40 | 0.2445 | 0.6980 | 0.3100 | MB27 | 0.40 | 0.2160 | 0.5850 | 0.3200 |
| MB16 | 0.40 | 0.1945 | 0.7870 | 0.3000 | MB43 | 0.40 | 0.2815 | 0.8660 | 0.3000 |
| MB38 | 0.40 | 0.2980 | 0.8480 | 0.4000 | MB22 | 0.20 | 0.0825 | 0.5570 | 0.2600 |
| MB13 | 0.20 | 0.1005 | 0.5760 | 0.1800 | MB28 | 0.20 | 0.0700 | 0.5050 | 0.2200 |
| MB14 | 0.20 | 0.0645 | 0.5770 | 0.2200 | MB42 | 0.20 | 0.1480 | 0.6620 | 0.2000 |
| MB29 | 0.20 | 0.0980 | 0.6870 | 0.2400 | | | | | |
| MB37 | 0.20 | 0.1175 | 0.6010 | 0.2500 | | | | | |

**Table 2.** Dependence models of increased volume, expansion and deepening of the borehole on the amount of explosive charge.

| Model | Basic | Linearized |
|---|---|---|
| Exponential | $y = a*e^{(b*x)}$ | $\ln(y) = b*x + \ln(a)$ |
| Logarithmic | $y = a*\ln(x) + b$ | $e^{((y - b)/a)} = x$ |
| Modified logarithmic | $y = a*\ln(x + b)$ | $e^{((y/a))} = x + b$ |
| Power | $y = a*x^b$ | $(y/a)^{(1/b)} = x$ |

Since the borehole has its initial volume, diameter, and depth, the values of the initial volume ($V_{rc0}$), diameter ($D_{re0}$), and depth of expansion ($L_{re0}$) are added to the experimental data. These points in scatter plot have coordinates (0, $V_{rc0}$), (0, $D_{re0}$), and (0, $L_{re0}$) (Figures 10–12).

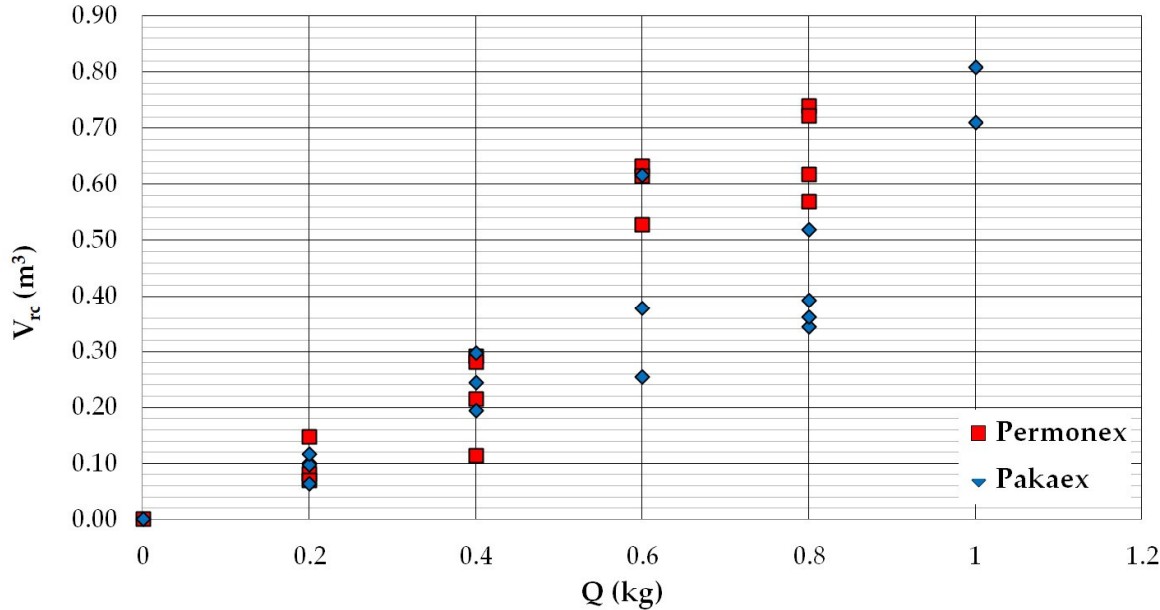

**Figure 10.** The dependence of the resulting volume of expansion $V_{rc}$ (m³) on the mass of explosive charge $Q$ (kg), for Permonex V19 and Pakaex.

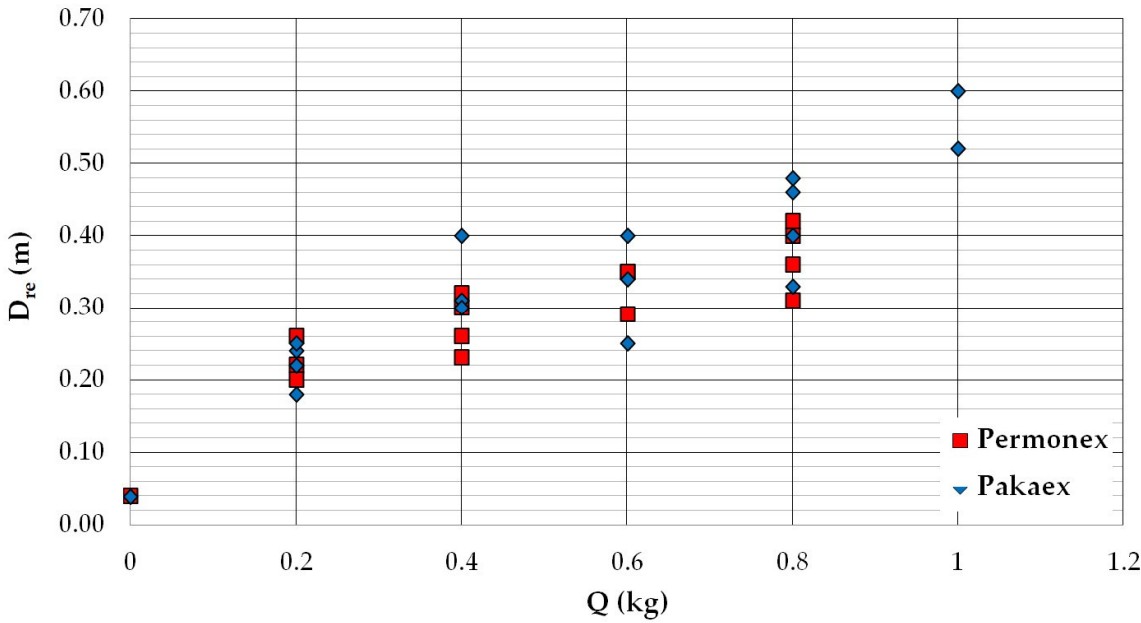

**Figure 11.** The dependence of diameter of enlargement $D_{re}$ (m) on the mass of explosive charge Q (kg), for Permonex V19 and Pakaex.

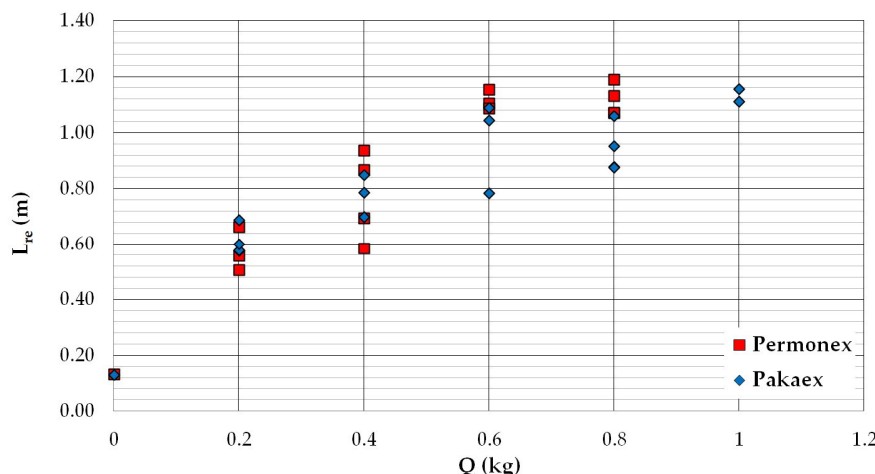

**Figure 12.** The dependence of the resulting deepening $L_{re}$ (m) on the mass of explosive charge Q (kg), for Permonex V19 and Pakaex.

However, a logarithmic model in its original form cannot be applied in this case. Namely, if Zero is included as the value of an independent variable, the coordinates of these points are $(0, -\infty)$. For this reason, parameter b should be appended to the function argument:

$$x \rightarrow x+b \Rightarrow f(x) \rightarrow f(x+b), \tag{1}$$

and thus, formed a modified logarithmic model which, like other models, has two parameters, Table 2.

This analysis includes three blasting performance indicators, two databases (basic and supplemented by baseline volume, expansion and deepening), three functions and two types of explosive. In this way, a total of 36 models were formed, and the values of their parameters were determined by the least square method, Table 3.

**Table 3.** Model parameters values.

| | | Volume | | | | Expansion | | | | Deepening | | | |
| | | Pakaex | | Permonex V19 | | Pakaex | | Permonex V19 | | Pakaex | Permonex V19 | Pakaex | Permonex V19 |
| | | Basic | Bs.+ nul.v. | Basic | Bs.+ nul.v. | Basic | Bs.+ nul.v. | Basic | Bs.+ nul.v. | Basic | Bs.+ nul.v. | Basic | Bs.+ nul.v. |
|---|---|---|---|---|---|---|---|---|---|---|---|---|---|
| Exp. | a | 0.09 | 0.09 | 0.10 | 0.10 | 0.57 | 0.51 | 0.52 | 0.46 | 0.19 | 0.17 | 0.20 | 0.17 |
| | b | 2.03 | 2.10 | 2.38 | 2.47 | 0.68 | 0.84 | 1.02 | 1.20 | 1.04 | 1.16 | 0.79 | 1.03 |
| Log. | a | 0.31 | 0.93 | 0.44 | 1.24 | 0.29 | 1.37 | 0.43 | 1.77 | 0.16 | 0.66 | 0.11 | 0.54 |
| | b | 0.57 | 0.92 | 0.75 | 0.90 | 1.07 | 1.31 | 1.23 | 1.15 | 0.47 | 1.16 | 0.39 | 1.24 |
| Pow. | a | 0.66 | 0.66 | 0.94 | 0.94 | 1.09 | 1.09 | 1.30 | 1.30 | 0.50 | 0.50 | 0.40 | 0.40 |
| | b | 1.20 | 1.20 | 0.21 | 1.32 | 0.35 | 0.35 | 0.51 | 0.51 | 0.53 | 0.53 | 0.37 | 0.37 |

Since models are nonlinear, they need to be linearized to determine the confidence line, Table 2. Subsequently, the Pearson coefficient (r) value for each linearized model was determined and the t-test (Table 4) was applied. The level of significance is the same for all models and is 5%. For a model to be reliable, the resulting T value for the linearized model must be greater than the critical value ($t_\alpha$).

**Table 4.** T-test results.

| Linearized Models | | PAKAEX | | | | PERMONEX V19 | | | |
| | | Basic | | Supplemented | | Basic | | Supplemented | |
| | | r | t | r | t | r | t | r | t |
|---|---|---|---|---|---|---|---|---|---|
| Volume | Exp. | 0.9054 | 7.979 | 0.7614 | 4.549 | 0.9125 | 7.727 | 0.9125 | 8.043 |
| | Log. | 0.8458 | 5.932 | 0.8543 | 6.366 | 0.9299 | 8.758 | 0.9418 | 10.101 |
| | Pow. | 0.8893 | 7.276 | 0.9088 | 8.436 | 0.9369 | 9.284 | 0.9436 | 10.276 |
| Expansion | Exp. | 0.8754 | 6.776 | 0.7522 | 4.421 | 0.8746 | 6.249 | 0.8269 | 5.302 |
| | Log. | 0.8602 | 6.311 | 0.8866 | 7.424 | 0.8935 | 6.893 | 0.9225 | 8.536 |
| | Pow. | 0.7308 | 5.816 | 0.8624 | 6.598 | 0.8944 | 6.927 | 0.9182 | 8.358 |
| Deepening | Exp. | 0.863 | 6.394 | 0.787 | 4.934 | 0.913 | 7.727 | 0.936 | 9.604 |
| | Log. | 0.848 | 5.984 | 0.890 | 7.572 | 0.916 | 7.883 | 0.935 | 9.466 |
| | Pow. | 0.846 | 5.937 | 0.872 | 6.889 | 0.893 | 6.889 | 0.896 | 7.275 |
| df | | 14 | | 15 | | 12 | | 13 | |
| t($\alpha$) | | 1.761 | | 1.753 | | 1.782 | | 1.771 | |

A $R^2$ value is specified for each model (Table 5). However, since the models differ from each other by quantity of data (N) on which, basis of parameters are defined, Akaike's Information Criteria (AIC) was used:

$$AIC = N*ln(SS/N) + 2K, \tag{2}$$

where: SS - sum of squared residual, K - number of parameters.

Since N is not much larger than K, corrected AIC ($AIC_c$) is used:

$$AIC_c = AIC + (2K(K + 1))/(N-K + 1), \tag{3}$$

The Obtained AIC and $AIC_c$ values for each model are listed in Table 5. The above equation shows that the $AIC_c$ value increases with the increase of N, SS and K, and the best model is considered the one where the $AIC_c$ value is the lowest.

**Table 5.** The values of the comparison measures. Blue is the lowest $AIC_c$ value for the measured data. Red is the lowest $AIC_c$ value for the measured data + $V_{rc0}$, $D_{re0}$ and $L_{re0}$

| | | Volume | | | | Expansion | | | | Deepening | | | |
| --- | --- | --- | --- | --- | --- | --- | --- | --- | --- | --- | --- | --- | --- |
| | | Pakaex | | Permonex V19 | | Pakaex | | Permonex V19 | | Pakaex | | Permonex V19 | |
| | | Measured | Me.+ $V_{rc0}$ | Measured | Me.+ $V_{rc0}$ | Measured | Me.+ $D_{re0}$ | Measured | Me.+ $D_{re0}$ | Measured | Me.+$L_{re0}$ | Measured | Me.+ $L_{re0}$ |
| N | | 16 | 17 | 14 | 15 | 16 | 17 | 14 | 15 | 16 | 17 | 14 | 15 |
| Exponential | $R^2$ | 0.78 | 0.80 | 0.82 | 0.83 | 0.74 | 0.70 | 0.75 | 0.76 | 0.77 | 0.77 | 0.71 | 0.70 |
| | $R^2_{adj}$ | 0.75 | 0.77 | 0.78 | 0.81 | 0.70 | 0.66 | 0.71 | 0.72 | 0.73 | 0.74 | 0.66 | 0.65 |
| | SS | 0.16 | 0.17 | 0.15 | 0.16 | 0.15 | 0.32 | 0.20 | 0.33 | 0.05 | 0.07 | 0.02 | 0.04 |
| | AIC | −69.35 | −74.12 | −59.37 | −63.95 | −70.45 | −63.56 | −55.42 | −53.34 | −88.58 | −89.59 | −90.24 | −85.83 |
| | $AIC_C$ | −68.55 | −73.37 | −58.44 | −63.09 | −69.65 | −62.81 | −54.49 | −52.14 | −87.78 | −88.84 | −89.32 | −84.97 |
| Logarithmic | $R^2$ | 0.72 | 0.78 | 0.84 | 0.87 | 0.77 | 0.80 | 0.81 | 0.87 | 0.69 | 0.80 | 0.70 | 0.79 |
| | $R^2_{adj}$ | 0.67 | 0.74 | 0.81 | 0.84 | 0.74 | 0.77 | 0.77 | 0.84 | 0.64 | 0.77 | 0.64 | 0.76 |
| | SS | 0.21 | 0.19 | 0.13 | 0.13 | 0.13 | 0.22 | 0.16 | 0.18 | 0.07 | 0.06 | 0.02 | 0.03 |
| | AIC | −69.05 | −72.10 | −61.22 | −66.98 | −72.78 | −70.13 | −58.93 | −62.01 | −83.95 | −91.55 | −89.58 | −91.42 |
| | $AIC_C$ | −68.25 | −71.35 | −60.30 | −66.13 | −71.98 | −69.38 | −58.01 | −61.16 | −83.15 | −90.80 | −88.66 | −90.57 |
| Power | $R^2$ | 0.77 | 0.80 | 0.87 | 0.89 | 0.78 | 0.86 | 0.81 | 0.87 | 0.72 | 0.80 | 0.71 | 0.85 |
| | $R^2_{adj}$ | 0.74 | 0.77 | 0.85 | 0.87 | 0.74 | 0.84 | 0.77 | 0.85 | 0.68 | 0.78 | 0.66 | 0.83 |
| | SS | 0.17 | 0.17 | 0.11 | 0.11 | 0.13 | 0.15 | 0.16 | 0.17 | 0.06 | 0.06 | 0.02 | 0.02 |
| | AIC | −68.70 | −74.28 | −64.26 | −70.17 | −72.92 | −76.67 | −59.01 | −62.97 | −85.92 | −92.13 | −90.32 | −96.80 |
| | $AIC_C$ | −67.90 | −73.53 | −63.34 | −69.32 | 72.12 | −75.92 | −58.09 | −62.12 | −85.12 | −91.38 | −89.41 | −95.94 |

## 6. Results and Discussion

The dependency plots shown in Figures 10 and 11 clearly indicate that the resulting volume, $V_{rc}$ ($m^3$) and diameter, $D_{re}$ (m) of expansion at the same mass of explosive charge Q (kg), is greater after activation of Permonex V19 than of Pakaex explosive. Conversely, the deepening result, $L_{re}$ (m) at the same mass of explosive charge Q (kg), is greater when using Pakaex than Permonex V19 explosive (Figure 12).

With regard to the abovementioned explanation, it is concluded that the specific blasting properties, velocity of detonation, brisance and the work capability of explosives, and the volume and specific explosion pressure, detonation pressure, and the density of the explosive, have a decisive impact on the results obtained when the spherical cavity blasting is performed.

For explosives used during the research, following manufacturer's data were utilized: the density of explosives, the velocity of detonation, the explosion temperature, and the volume of gases.

For quality conclusion, it would be necessary to carry out an additional research on which blasting properties have a greater impact on the resulting volume and diameter of the expansion, and which one on the deepening of the borehole. For this purpose, it is necessary to know the detonation process, whereby it seems that the most important property is the detonation front [22,23].

What is undoubtedly proven is that detonation of the same mass of Permonex V19, which has a higher density and velocity of detonation, produce greater volume and diameter than the detonation of Pakaex which has a lower density and velocity of detonation.

The resulting parameter values were included in the linearized models and their reliability was determined. T-test results are listed in Table 4. All the derived t-values are greater than the critical values. It is therefore acceptable to conclude that all proposed models are reliable and can be mutually compared.

Determination coefficient values ($R^2$) range from 0.69 to 0.89 (Table 5). Based on these results, it can be concluded that all proposed models successfully fit into experimental data. However, because of the difference between the numbers of parameters in the models, the AIC or $AIC_c$ value (Table 5), is relevant for mutual comparison of the models. Namely, in two cases (Pakaex, expansion, logistical model and Pemonex V19, expansion, exponential model) obtained $R^2$ values are higher in the supplemented models, which means that these models are better than the basic ones. However, the obtained AICc values are lower for basic models.

It should be noted that in most cases the $AIC_c$ value gives exactly the opposite conclusion. Namely, if the basic and supplemented models are mutually compared for the same indicator, the same function and the same explosive, 18 pairs of models are obtained. In 77.78% cases (14 out of 18), the $AIC_c$ value is lesser for the supplemented model (Table 5).

In addition, the basic and complementary models for the same indicator and the same explosive (regardless of function) are mutually compared. In this way, six cases are obtained. Figures 13–15 show plots of mathematical models in which the lowest $AIC_c$ values are obtained for both types of explosive.

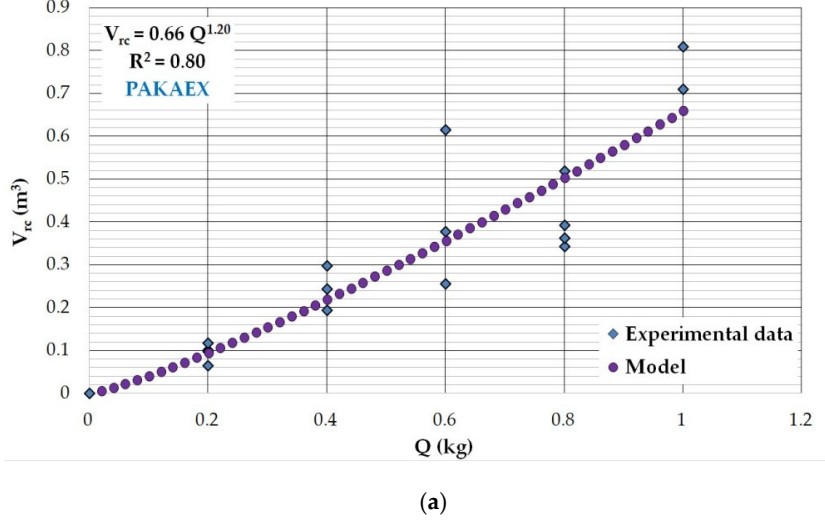

(**a**)

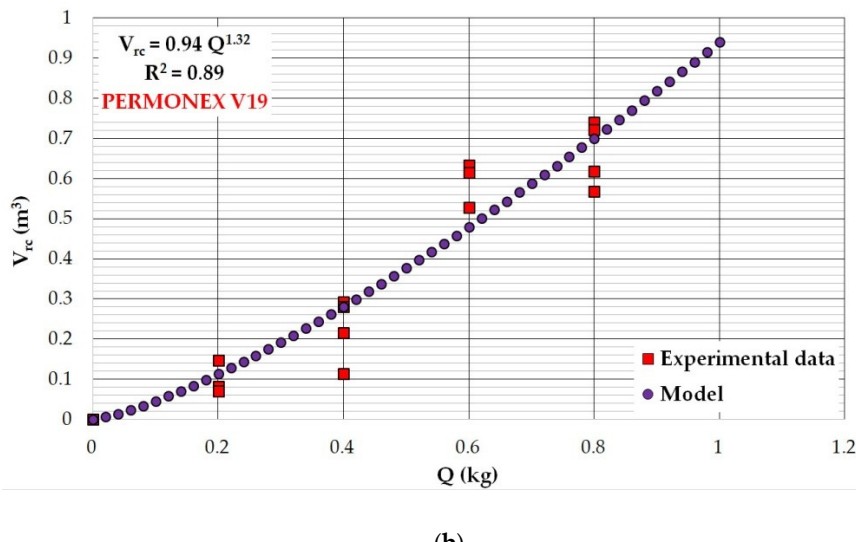

(**b**)

**Figure 13.** (**a**) The mathematical model plots of the lowest $AIC_c$ values for the resulting expansion volume $V_{rc}$ (m$^3$) for Pakaex. (**b**) The mathematical model plots of the lowest $AIC_c$ values for the resulting expansion volume $V_{rc}$ (m$^3$) for Permonex V19.

In all these cases the lowest $AIC_c$ value is obtained for the supplemented model. This fact confirms the justification for adding the constant to the function argument as a new model parameter.

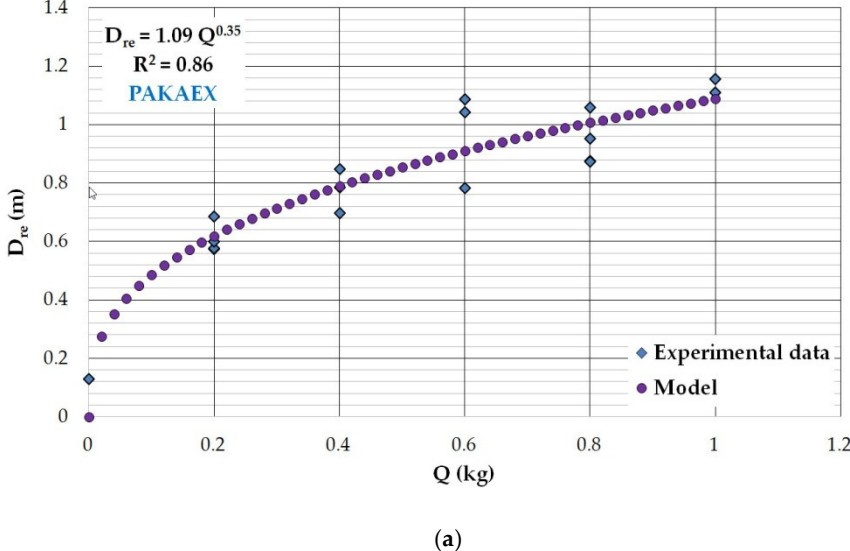

(**a**)

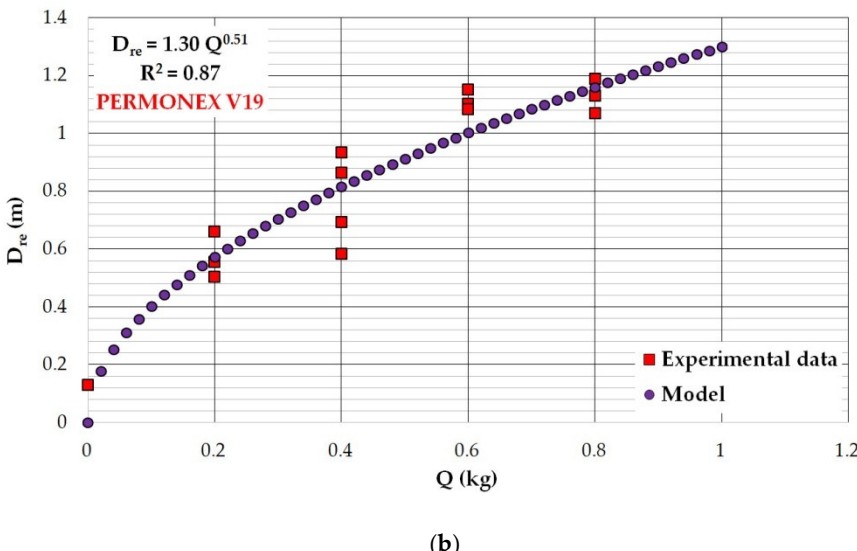

(**b**)

**Figure 14.** (**a**) The mathematical model plots of the lowest $AIC_c$ values for the expansion diameter $D_{re}$ (m) for Pakaex. (**b**) The mathematical model plots of the lowest $AIC_c$ values for the expansion diameter $D_{re}$ (m) for Permonex V19.

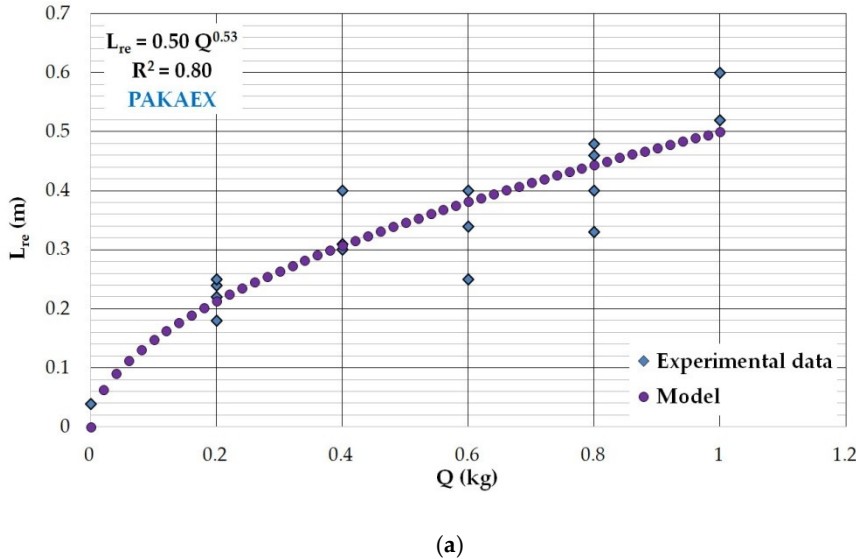

(**a**)

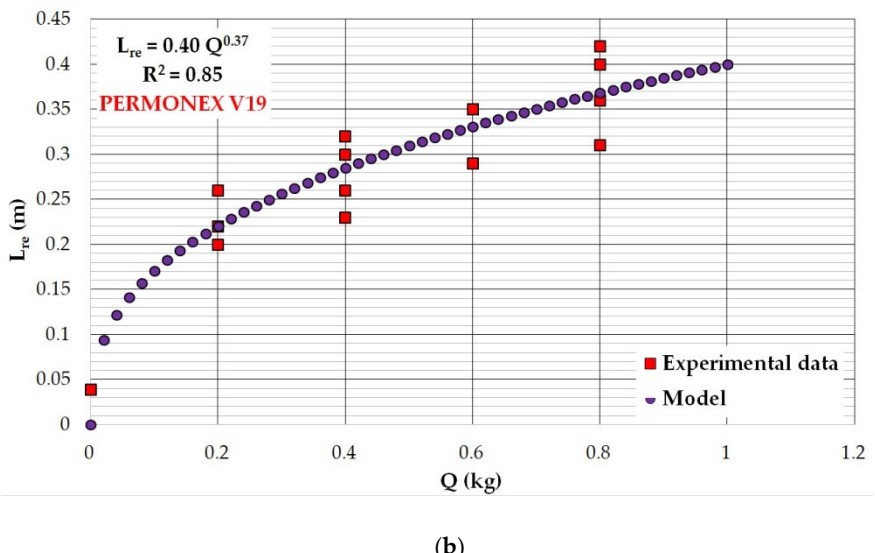

(**b**)

**Figure 15.** (**a**) The mathematical model plots of the lowest $AIC_c$ values for the resulting deepening $L_{re}$ (m) for Pakaex. (**b**) The mathematical model plots of the lowest $AIC_c$ values for the resulting deepening $L_{re}$ (m) for Permonex V19.

## 7. Conclusions

The main goal of this research was expanding the knowledge about the possibilities of using explosives in geotechnical practice. That refers particularly to the spherical cavity blasting in clay on different depths by activating a specific type and mass of explosive charge. The practical application of obtained results is the compaction of coherent clay soils using explosive charges for stabilization of slopes by anchoring, different surface and underground structures, embedding of constructive elements for anchoring foundations and retaining walls, stabilization of different structures, such as overhead transmission towers, tunnels, etc.

The aim of the research was to determine the influence and optimization of specific parameters during spherical cavity blasting process with the intention of obtaining the desired volume of the cavity.

One of the most important results of this research is determination of the volume of obtained cavities in clay soil during blasting in the boreholes of specific profile. For this purpose, a unique

system, that integrates RTK GNSS method, was developed at the Faculty of Geotechnical Engineering, by applying online transformation parameters from CROPOS system. The developed application calculates the coordinates of the borehole based on known E and N coordinates (obtained for the GPS) and height H (in this case the depth of the borehole) obtained with a Heavy Duty GeoVision Borehole Camera. In order to calculate all the coordinates of the boreholes on a specific depth, a laser (EDS-C) was used to determine the distance between the depth camera and the wall of the borehole. For the purpose of calculating the volume of the cavity, application "Bušotine" was developed, which represents a significant step forward and innovation in the field of geotechnical engineering. Apart from calculating the volume, the application also maps 2D and 3D models of the spherical cavity. In addition, it is compatible with other CAD tools, making it easier to check and add details to the 3D model.

Finally, in this paper, a clear correlation is established between the spherical cavity volume and the blasting charge mass and type of explosive used, as well as its detonation and blasting parameters. This makes a solid base for the development of model of spherical cavity blasting in different types of soils and rocks. This can be used in geotechnical practice for stabilization and improvement of soil and rock characteristics.

It should be emphasized that, if basic and supplemented model are mutually compared for same indicators, same function, and same explosive, 18 pairs of models are obtained. In 77.78% cases (14 of 18) AICc value is lesser for the supplemented model (Table 5). This justifies the addition of the constant b to the function argument as a new model parameter.

**Author Contributions:** Conceptualization, D.T.; formal analysis, I.K.; investigation, D.T.; methodology, I.K.; project administration, S.S.; resources, D.T.; software, I.K.; supervision, D.T.; visualization, S.S.; writing—original draft, D.T.; writing—review and editing, S.S.

**Funding:** This research received no external funding.

**Acknowledgments:** Publication process is supported by the Development Fund of the Faculty of Mining, Geology and Petroleum Engineering, University of Zagreb.

**Conflicts of Interest:** The authors declare no conflict of interest.

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
