# Peer review of "Dependence Models of Borehole Expansion on Explosive Charge in Spherical Cavity Blasting"

_geosciences, doi:10.3390/geosciences9090383_

Round 1

Reviewer 1 Report

I have not comments

Author Response

We would like to thank Reviewer 1 for kind revision.

Reviewer 2 Report

Figure 4: the first part of the figure description is missing.

line 63 the explosives types are not defined in details, properties of explosives are missing, as well the properties of explosives used in testing are missing (velocity of detonation, volume, density etc.)

line 262 attention for the capital letter: For This purpose, it is necessary 
same line 293 In Addition, the basic 

Author Response

We would like to thank Reviewer 2 on comments and suggestions, which are all implemented in revised article as it follows:

Point 1: Moderate English changes required. 

Response 1: Changes were made throughout the article (please see comments within article)

Point 2: Figure 4: the first part of the figure description is missing.

Response 2: The word “Location” added (please see comment line 121)

Point 3: line 63 (new line 65) the explosives types are not defined in details, properties of explosives are missing, as well the properties of explosives used in testing are missing (velocity of detonation, volume, density etc.).

Response 3: Properties of explosives are added (please see comment line 65-67)

Point 4: line 262 (new line 269) attention for the capital letter: For This purpose, it is necessary same line 293 (new line 299) In Addition, the basic

Response 4: Rectified (please see comment lines 269 and 299)

Reviewer 3 Report

Figure 12 volume (Vrc) should be in m3.

The authors presented interesting research results that will definitely be implemented in the future in geotechnical processes.
It is puzzling, however, that there are very large differences in the volume (Vrc) of the cavities obtained when detonating the same mass of explosive charge, and even obtaining a larger volume (Vrc) from smaller charges. Please, try to explain the reasons for this phenomenon.
To make the test more representative, I suggest increasing the number of shots for a given mass of explosives in the future.

Can the range of soil parameters to which the described method can be applied be determined based on the tests carried out?

Was the same stemming length used for each explosive mass? If so, why? Too short stemming with a larger mass of explosive may cause smaller cavern volumes.

On what basis was the 0.5 m stemming selected? What is the limit value of the stemming length at which part of the explosive detonation energy is lost to its ejection from the borehole?

Would closing the borehole entirely with a stemming, and then its mechanical removal, would not give better results in the form of a larger cavern volume?

Author Response

We would like to thank Reviewer 3 on comments and suggestions, which are all implemented in revised article as it follows:

Point 1: Figure 12 volume (Vrc) should be in m3.

Response 1: Rectified (please see comment line 220/221)

Point 2: It is puzzling, however, that there are very large differences in the volume (Vrc) of the cavities obtained when detonating the same mass of explosive charge, and even obtaining a larger volume (Vrc) from smaller charges. Please, try to explain the reasons for this phenomenon.

Response 2: Large differences in the volume of the cavity obtained at the same mass of explosives, even in smaller quantities derive mainly from the properties of the explosives used. The Permonex V19, generates a larger volume of gases and has higher detonation pressures than that of the PAKAEX explosive type. In this way, a compaction in clay soil is increased, resulting in a larger volume of expansion.

Point 3: To make the test more representative, I suggest increasing the number of shots for a given mass of explosives in the future.

Response 3: Thank you for the suggestion, we will do that in the future

Point 4: Can the range of soil parameters to which the described method can be applied be determined based on the tests carried out?

Response 4: We got some preliminary results. However, since core drilling is expensive, it has been done only on 3 locations. Therefore, we did not want to include it in this article.

Point 5: Was the same stemming length used for each explosive mass? If so, why? Too short stemming with a larger mass of explosive may cause smaller cavern volumes.

On what basis was the 0.5 m stemming selected? What is the limit value of the stemming length at which part of the explosive detonation energy is lost to its ejection from the borehole?

Response 5: We would like to apologize for mistake in our article. The stem during the test was 0.3 m and 0.5 m, depending on quantity of explosive in the borehole, not to allow its ejection from it, which, we fully agree with your comment, would result in partial detonation energy loss.

Rectified (please see comment line 176-179)

Point 6: Would closing the borehole entirely with a stemming, and then its mechanical removal, would not give better results in the form of a larger cavern volume?

Response 6: We tried it. However, we encountered problem with mechanical removal of the large stem. During removal with drill rig, borehole partially collapsed along with stem material into cavern, making precise measurement of cavern impossible. Therefore, we did not acquire any conclusive results to support or reject this statement.
